# Comparison of Two Bovine Commercial Xenografts in the Regeneration of Critical Cranial Defects

**DOI:** 10.3390/molecules27185745

**Published:** 2022-09-06

**Authors:** Carlos Humberto Valencia-Llano, Diego López-Tenorio, Marcela Saavedra, Paula A. Zapata, Carlos David Grande-Tovar

**Affiliations:** 1Grupo Biomateriales Dentales, Escuela de Odontología, Universidad del Valle, Calle 4B # 36-00, Cali 76001, Colombia; 2Grupo de Polímeros, Facultad de Química y Biología, Universidad de Santiago de Chile (USACH), Santiago 9170020, Chile; 3Grupo de Investigación de Fotoquímica y Fotobiología, Universidad del Atlántico, Carrera 30 Número 8-49, Puerto Colombia 081008, Colombia

**Keywords:** bone substitutes, collagen membranes, critical size defects, xenografts

## Abstract

Autologous bone is the gold standard in regeneration processes. However, there is an endless search for alternative materials in bone regeneration. Xenografts can act as bone substitutes given the difficulty of obtaining bone tissue from patients and before the limitations in the availability of homologous tissue donors. Bone neoformation was studied in critical-size defects created in the parietal bone of 40 adult male Wistar rats, implanted with xenografts composed of particulate bovine hydroxyapatite (HA) and with blocks of bovine hydroxyapatite (HA) and Collagen, which introduces crystallinity to the materials. The Fourier-transform infrared spectroscopy (FTIR) analysis demonstrated the carbonate and phosphate groups of the hydroxyapatite and the amide groups of the collagen structure, while the thermal transitions for HA and HA/collagen composites established mainly dehydration endothermal processes, which increased (from 79 °C to 83 °C) for F2 due to the collagen presence. The xenograft’s X-ray powder diffraction (XRD) analysis also revealed the bovine HA crystalline structure, with a prominent peak centered at 32°. We observed macroporosity and mesoporosity in the xenografts from the morphology studies with heterogeneous distribution. The two xenografts induced neoformation in defects of critical size. Histological, histochemical, and scanning electron microscopy (SEM) analyses were performed 30, 60, and 90 days after implantation. The empty defects showed signs of neoformation lower than 30% in the three periods, while the defects implanted with the material showed partial regeneration. InterOss Collagen material temporarily induced osteon formation during the healing process. The results presented here are promising for bone regeneration, demonstrating a beneficial impact in the biomedical field.

## 1. Introduction

Bone regeneration is a technique aimed at stimulating the neoformation of bone tissue in places where it has been lost. Usually, materials from the same patient (autologous bone) or other sources such as homologous tissues (from human donors), xenologous (originated in other species), and alloplastics (synthetic) are used for bone regeneration [1]. The ideal materials must present at least one of these qualities: they should be osteoconductive (provide support), osteoinductive (contain biological factors that incite cell differentiation), osteogenic (contain cells that carry out bone formation) and promote osseointegration (ability to join the bone tissue through a direct union, with no fibrous encapsulation). Autologous bones are the unique materials with these four qualities, considered the gold standard for bone regeneration [2]. A new quality incorporated into bone substitutes is osteopromotion, or the property of promoting integration to the rest of the tissue through the incorporation of some non-inducing biological agents [3]. The use of xenografts as bone substitutes arose from the difficulty of having autologous bone and the limited availability of tissue and organ donors [4]. It is recognized that these materials are only osteoconductive [5,6,7]. However, they might introduce an antigenic response and some native properties might be lost due to the chemical extraction [2]. Difficulties in controlling the presence of prions were also reported [5,8], although this biological risk is considered minimal coming from a deproteinated and decellularized bone matrix [9,10]. Other adverse effects mentioned in the literature are osseointegration problems such as encapsulation at the graft site that persists after several years, slow resorption [6], and complications such as soft tissue fenestration and cyst formation that would warrant further study [11].

The addition of collagen to xenografts was performed to improve the biological properties. Collagen is a main extracellular bone matrix component with osteopromoting capacity. For that reason, adding xenografts improves the graft’s ability for osseointegration with the biological environment. Type I collagen is the most abundant protein in bone tissue. It is directly involved in the mineralization of the matrix and it is resorbable [12]. Still, it is also desirable as a substrate for anchoring cells due to its ability to bind through integrins [13]. Additionally, positive effects on healing have been reported by incorporating it into compounds with calcium phosphates [14,15]. 

The critical defect size is an experimental design widely used to study the regenerative capacity of a material with application in bone tissue. It is based on making a surgical preparation such a large size that it cannot heal spontaneously during the experiment. However, if new bone formation occurs, it will not be more than 30%. A five millimeter preparation in the parietal bone that does not include the cranial sutures is considered of critical size [16,17]. 

Two xenografts were studied in our research based on bovine HA particles (InterOss) and another that includes collagen fibers to the HA particles (InterOss Collagen), based on previous biocompatibility studies [18]. In the literature review, we found very few studies with the biological characterization of these two products. In the case of InterOss, one was found to preserve dental alveoli in humans [19]. However, this research bases the results on the analysis of tomographic images. InterOss has also been implanted in small preparations in rat condyles [20]. An investigation was found where InterOss was applied to critical size defects in the alveolar bone of beagle dogs [21]. 

In the case of InterOss Collagen, a study on a rabbit femur was found [22], but it also focused on implantations in non-critical size defects. In our investigation, the two materials were applied to critical size defects in the parietal bone of the Wistar rat. This preparation has the particularity that the cranial bones of the rat have a straightforward vascular system without presenting the characteristic “osteons-like” of the Haversian system [23]. In this way, the results observed in the cranial model can be mainly attributed to the properties of the implanted materials since it is guaranteed that spontaneous healing will not occur during the experiment, a recognized quality for defects of critical size [16]. Thus, any bone regeneration will be attributed to xenografts, not a spontaneous healing process. 

## 2. Results and Discussion

In this work, we compared the critical size defects regeneration capacity of two xenografts and a collagen membrane. The first xenograft is a particulate bovine apatite (F1, InterOss). This particulate xenograft simulates a human bone due to a highly porous, thermal-stable and osteoconductive structure, which also facilitates the osteoconduction and osteoinduction, and regeneration of the human bone within the pores [24,25]. According to the manufacturer, F1 presents a surface area of 88.2 ± 0.015 m^2^g^−1^, similar to human bone (50–100 m^2^ g^−1^), with abundant Ca (63 wt.%) and P (33 wt.%) [24]. 

The second xenograft is a collagen and HA particles composite (F2, InterOss Collagen) with an irregular distribution of high macro and micropores, cuboid morphology, and a large surface area (77.0 ± 0.2 m^2^ g^−1^), also in the human-being range [22]. The presence of collagen avoids the collapse and aggregation of particles or the low collagen resistance from membranes, which introduces discomfort and risk of infection [26]. 

The third formulation corresponds to a porcine collagen membrane (F3), InterCollagen Guide highly porous, fibrous, and stretchable with fluid absorption capacity. However, these membranes degrade quickly, with a high infection risk for the patient. We previously characterized a different batch of the formulations and tested the preliminary biocompatibility in vivo and in vitro using subdermal implantations in biomodels [18]. Our results demonstrated high biocompatibility in subdermal implantations of the materials. However, the in vivo analysis for cranial critical size defects in Wistar rats has not yet been reported. We first chemically characterized the formulations, analyzed the crystallinity and thermal transitions, characterized the morphology of the materials, and analyzed the impact of the material’s properties in the in vivo bone implantations. 

### 2.1. Composite Characterization

#### 2.1.1. Fourier-Transform Infrared Spectroscopy (FT-IR)

Despite the xenografts and the membrane corresponding to a different batch, we expected similar FT-IR bands for the main functional groups of HA (F1), HA/collagen (F2), and collagen (F3). Figure 1A shows the main functional groups of the composites prepared in our research. F1 main bands correspond to CO_3_^−2^ from HA (962 cm^−1^, 1416 cm^−1,^ and 1456 cm^−1^), as evidenced in Figure 1B, the expanded region of 600 to 1700 cm^−1^ [24,27]. Furthermore, the orthophosphates (PO_4_^−3^) bands were evidenced at 562 cm^−1^, 601 cm^−1^, and 1025 cm^−1^ (Figure 1B) [22,28]. 

Secondly, F2 presented bands of bovine HA similar to F1 due to the 90 wt. % HA content. However, collagen peaks are also patent due to the low collagen content of F2 (10 wt.%). The bands at 3343 cm^−1^ and 1651 cm^−1^ are attributed to N–H and C=O of amides I. Moreover, peaks at 562 cm^−1^, 601 cm^−1^, 962 cm^−1^, 1025 cm^−1^, 1416 cm^−1,^ and 1456 cm^−1^ related to CO_3_^−2^ and PO_4_^−3^ from HA are like the peaks of F1 (Figure 1B). 

On the other hand, F3 presented typical collagen bands, since the membrane is mainly produced from bovine collagen [29]. The peaks at 3298, 3075 (N–H amide I groups), 2949 cm^−1^ (CH_2_), 1644 cm^−1^ (C=O amide I), and 1548 cm^−1^ (N–H and C–N amides II), 1210 and 1236 cm^−1^ were observed [29,30]. Additionally, we observed the peaks at 1210 and 1236 cm^−1^ corresponding to N–H, C–N, and C–H of amide III [29,31]. Finally, traces of CO_3_^−2^ generated the 1100 cm^−1^ and 595 cm^−1^ peaks [32]. All these observations are similar to our previous report [18].

#### 2.1.2. Thermal Characterization

##### Differential Scanning Calorimetric (DSC) 

We studied the thermal transitions from the xenografts and F3 membranes using differential scanning calorimetry (DSC) (Figure 2). F1 shows an endothermic dehydration transition centered at 79 °C. On the other hand, F2 presents a dehydration peak at a higher temperature (83 °C), probably influenced by the change in orientation of the collagen chains, from an organized structure of triple-helical helix to a disorganized one, thanks to the rupture of the covalent interactions between the polymeric chains [22,33]. For F2, unlike F1, we observed a second peak at 223 °C corresponding to the helical structure transitions and the beginning of protein denaturation [22,34]. Finally, F3 exhibited a higher and more intense than F1 and F2 dehydration endothermic peak at 89 °C due to the presence of more hydrophilic groups (mainly hydroxyl and amine groups) available to retain more water than the xenografts F1 and F2. Furthermore, water penetration to F3 fibers is easier than F1 and F2 due to the HA compact structure observed by SEM. F3 also exhibited the peak at 223 °C attributed to the transition from the helical structure in collagen to a disordered structure. After 250 °C, we can observe the beginning of the pyrolysis of collagen chains only for F3 [34,35]. All the results are according to our previously reported findings for thermal studies of xenografts and collagen membranes from a different batch [18]. 

#### 2.1.3. X-ray Diffraction Studies for the Xenografts 

XRD analysis helps analyze the chemical composition, phases, and crystallinity of inorganic structures [28]. Figure 3 shows the diffraction pattern of the three formulations from our study. 

According to Figure 3, the crystallization for F1 and F2 had a similar diffraction pattern. The angles 20° < 2θ < 50° show a prominent peak at 32°, corresponding to a crystalline bovine HA [36]. The diffraction peaks for F1 and F2 correspond to HA structure, with hkl values related to 002, 102, 210, 211, 112, 300, 202, 310, 222, 213, 321, and 004 planes of the HA, demonstrating that InterOss (F1) and InterOss Collagen (F2) had a crystalline ordered structure of HA [28]. The crystallinity study of the materials is essential to understanding biosorption, which is dependent upon tissue repairing. Moreover, the calcium and phosphate amounts in the calcium phosphates also affect solubility and crystallinity, since a higher crystallinity will decrease the material’s solubility [37]. The previous statement supports that F2 with a 10 wt.% content of collagen will have a lower crystallinity and be more soluble and resorbable. This observation is related to the higher endothermic peak due to the dehydration for F2 as compared to F1 observed in the DSC analysis, indicating more water content in F2, contributing to a higher degradation/resorption in the biological fluids, as observed in the in vivo implantations. Figure 3 also shows the XRD pattern for the F3 (InterCollagen Guide) membranes, showing specific patterns for collagen XRD, with characteristic peaks centered at 2θ = 8°, 16°, and 23° for bovine collagen structures, very different from the F1 and F2 patterns [38]. The peak centered at 8° is related to the triple-helical structure of the collagen chains and indicates the lateral intermolecular chain distance [39,40]. The peak at 16° is related to the amorphous collagen structure from the membranes. The peak centered at 23° might be attributed to a helical structure of the Collagen used for the membrane preparation [41]. 

#### 2.1.4. Morphology Studies of Xenografts 

We investigated the morphology of the xenografts (F1 and F2) and the porcine collagen membrane (F3) using the scanning electron microscopy (SEM) technique (Figure 4). F1 and F2 exhibited a heterogeneous structure with an irregular porosity distribution, previously observed for this kind of xenografts [22]. F1 had a granular form with diameters around 1.5–2.5 µm, as observed previously [28], with macropores (diameters higher than 100 µm), essential for new bone formation through blood vessel proliferation (Figure 4A–C). Incorporating collagen in F2 makes the structure more compact (Figure 4D–F); however, due to the biodegradable nature of the collagen fibers, porosity is expected to increase after introduction to the body. The porcine collagen membranes (F3) are flexible and contain a crosslinked structure, with several macro and microporosity useful for bone regeneration (Figure 4G–I). All the structures exhibited a mesopore and micropore structures, facilitating new bone formation through the blood vessels, nutrient, and cell waste material transport. 

### 2.2. Histological Analysis of Xenografts

#### 2.2.1. Results of Cranial Implantation of Control (Empty Defect)

The critical size experimental designs of 5 mm diameters in rat cranial bone have been considered an appropriate design to test bone substitutes [42]. Two bilateral defects are created in the design and one is left empty (without grafting) as a control [43].

Figure 5 shows the healing behavior of empty defects at 30, 60, and 90 days. Through the three observation periods, it can be seen that the intraosseous preparation left without an implant begins a healing process from a fibrous type of healing. Figure 5A shows the preparation occupied by a disorganized connective tissue. At 60 days, some nodules of bone formation are identified (yellow stars in Figure 5B), immersed in a fibrous network. After 90 days, the healing of the preparation appears more organized and with larger and more defined nodes of bone formation, with the presence of type I collagen fibers (Figure 5C,D).

When the nodules of bone formation are examined at 40× magnification, a newly formed tissue is found with the presence of osteocytic lacunae occupied by cells in a tissue with the presence of type III collagen, observed by Masson’s trichrome staining (Figure 5E).

Figure 5F shows how the preparation has healed at the expense of fragile tissue of a fibrous nature (blue arrows). It is also observed how the integration with the surrounding bone is also fibrous (yellow arrows), fulfilling what is expected for a defect of critical size in that there is no spontaneous regeneration of the defect [16].

The 5 mm defect in the cranial bone of the Wistar rat is one of the most used and is accepted as a defect of critical size [16]. The original critical size defect concept put forth by Schmitz and Hollinger proposed that the size of the critical defect does not regenerate in the entire life of the animal [44]. However, this concept was redefined by the Gosain group (2002) as the slightest intraosseous preparation that did not regenerate spontaneously during experimentation [45].

In this work, partial bone neoformation was found in control defects. In critical size designs, new bone formation can be found in minimal amounts and generally in the periphery of the defect [46]. Still, for the procedure to remain valid, it must be less than 30% of the volume of the defect in implantations from one to three months [16]. This study found no new bone formation after 30 days of implantation. At 60 days, some small, isolated bone nodules were observed; at 90 days, larger bone nodules were observed. Still, they continued to be sheltered and in the process of maturation without reaching 30% of the preparation volume (Figure 5).

#### 2.2.2. Results of Cranial Implantation of F1 (InterOss)

SEM images for InterOss (F1) samples, a material that preserves the characteristics of the cancellous bone of the tissue of origin, are observed. We observed some fragments with irregular edges, rough surfaces, and the presence of pores of different sizes (Figure 6A). Additionally, structures compatible with bone trabeculae with medullary cavities can be seen (white arrows Figure 6B).

In the spectral analysis by SEM/EDS (Figure 7), high percentages of phosphorus and calcium were found.

#### 2.2.3. Results of Cranial Implantation of F1 (InterOss)

Histological images of the preparations grafted with the F1 (InterOss) material are presented below. Thirty days after implantation, abundant material particles immersed in the type I and III collagen network are observed in an image reminiscent of intramembranous ossification (Figure 8A–C). Figure 8C is an image of the meningeal surface. Several structures compatible with fragments of the material integrated into the scar tissue are observed.

Figure 8D–F correspond to samples within 60 days of implantation. The surgical defect has been occupied by bone tissue in the maturation process with an abundant presence of collagen (Figure 8D). The persistence of some particles of the material and some areas on the periosteal surface are still occupied by undifferentiated tissue (white ovals); the SEM image of the meningeal surface also allows observing some particles incorporated into the newly formed tissue.

At 90 days of implantation, the area of the preparation occupied by a more organized bone tissue can be seen, but with a different degree of maturation. Figure 8G corresponds to the almost complete regeneration of the defect with the persistence of a central zone in the formation process (oval in Figure 8G). At higher magnification, the central area occupied by small material particles in the process of degradation/reabsorption can be seen (Figure 8H). Using the SEM technique, it is also possible to observe some particles on the meningeal surface of the preparation. 

Xenografts are usually slow-reabsorbing materials [47]. In this investigation, remnant particles of InterOss were found in all periods. However, a decrease in particle presence was observed towards the third month, mainly concentrated in the middle portion of the preparation.

The decrease in the presence of remaining particles throughout the experiment indicates that the material is biodegradable/resorbable. The results are similar to those shown for xenografts such as BioOss and Osteograft [46]. It has also been reported that other xenografts such as Bone-Fill and Gen-Ox with Bio-Oss can stimulate bone regeneration in critical defects, but with different results concerning material degradation [48].

#### 2.2.4. Results of Cranial Implantation of F2 (InterOss Collagen)

SEM images exhibited an amorphous mass where some structures corresponding to the InterOss particles and some fibers from collagen are observed. The surface looks rough and porous. With the EDS analysis, we found a high percentage of calcium and phosphorus due to the hydroxyapatite (HA) component and a critical nitrogen content attributable to the presence of the collagen protein (Figure 9)

The histological images of the InterOss Collagen implantations (Figure 10) show that at 30 days, there is an abundant presence of the material occupying the entire implanted area, very similar to that shown in the same period for the InterOss material. We can observe the particles immersed in the types I and III collagen network (Figure 10A,B). Employing SEM (Figure 10C), it is observed that the meningeal surface of the preparation is covered by scar tissue corresponding to some particles of the material (red arrows). Additionally, it is observed that new tissue is growing from the peripheral bone, as indicated by the white arrows in Figure 10C. In this figure, we can also follow the defect’s interface with the remaining bone, some structures corresponding to cells (white stars).

In implantations at 60 days of F2, some particles persist, but numerous nodes of bone formation are also observed (Figure 10D). An interesting finding is the presence of osteon-compatible structures in the newly formed tissue (yellow circles in Figure 10D,E). Central channels such as Haversian and osteocytic lacunae with cells distributed concentrically concerning the central canal (green arrows) are observed. It can also be seen that these structures are delimited by lines such as the cement lines that delimit the “osteons-like” (green arrows). The SEM image shows complete tissue formation after 60 days (indicated with the number 1 in Figure 10F), with only a tiny, uncovered area (number 2 in Figure 10F). In addition, the deposition of a new layer of tissue is beginning (number 3 in Figure 10F, highlighted by a rectangle) that increasingly resembles the periosteum that covers the bone remnant (number 4 in Figure 10F, highlighted by a rectangle).

In the 90 days, the histological image exhibits normal-appearing bone tissue with some areas of tissue in the maturation process (ovals in Figure 10G,H) and numerous osteocytic lacunae, indicating that grafted material was replaced by newly formed bone tissue. Through histological studies, it was impossible to observe the material’s particles. However, the SEM image shows what could correspond to some remaining material (red arrows Figure 10I), in addition to the appearance of the tissue covering most of the preparation material, such as the periosteum of the surrounding bone tissue.

“Osteons-like” play an essential role in the blood supply of bone tissue by allowing the entry and exit of blood vessels through Haversian canals. A remarkable finding in these analyses is that the “osteons-like” observed in the two months disappear after three months, with only signs of what could have been an osteon (circle in Figure 10H) where a line of cement surrounds a rounded structure but without a central channel and concentric gaps. In this case, it was observed that the “osteons-like” are present at 60 days but disappear at 90, leaving only a few traces. These findings indicate that the InterOsss Collagen (F2) material stimulates more significant angiogenesis with the formation of these structures, facilitating healing and new bone formation. However, once normal bone tissue is formed, the remaining “osteons-like” disappear, establishing the tissue architecture of the cranial bones of this species, which have a simple blood supply.

The results found show a clear difference between InterOss and InterOss Collagen. In the first 30 days, there are no significant differences in the biological behavior of the two materials. However, at the end of the implantation period, it is observed that new bone formation is more homogeneous for InterOss Collagen (F2) as the amount of noticeable residual material decreases. 

Vascularization is crucial for any bone regenerative process, since the necessary nutrients for healing are provided, and waste products are eliminated [49]. In the case of the critical size defect in rats, vascularization is compromised due to the essential concept of size and tissue type. The cranial bones of rats are considered to have simple vascularization because they do not present the formation of “osteons-like” [50].

“Osteons-like” are structures related to the vascularization of cortical bone and cancellous bone. The finding of primary “osteons-like” in a bone where they are generally not found is indicative of an adaptation made by the bone system to guarantee the presence of the necessary blood vessels during the healing process; from there, they disappear when a degree of maturity of the newly formed tissue is reached. The formation of “osteons-like” in the 60 days may be related to the presence of collagen in the biomaterial. This situation has been described in implantations with products with chitosan, another osteopromoting material [50].

The difference in the material reabsorption dynamics between the two products is due to the degradability of the grafted materials and the presence of collagen, which can stimulate the healing process. The degradability of bone xenografts is considered slow, but that of collagen is fast due to enzymatic reaction [51]. Combining a particulate xenograft with collagen fibers should have a synergistic effect. The synergistic impact would adjust the reabsorption of apatite particles and stimulate bone regeneration, as collagen is an essential component of the extracellular matrix and can promote cell differentiation, growth, adhesion, and mineralization [52], especially angiogenesis [53].

In a histological evaluation of sites grafted with a bone xenograft supplemented with collagen, an earlier ossification process was found than in the control sites, with more significant amounts of new bone formed [54]. Here, we found similar results of new bone formation, different from the control sites. The three months of implantation results showed the minimal presence of residual material in the middle surrounded by newly formed bone, which agrees with what has been reported for implantations of the same material in rabbits [22].

#### 2.2.5. Results of Cranial Implantation of F3 (InterCollagen Guide)

Collagen membranes have been widely used in the guided bone regeneration technique as a barrier that keeps epithelial cells away. In contrast, bone regeneration occurs in the sites of interest [55]. The barrier must remain in place and last long enough for the grafted material to be replaced by bone tissue [54]. The performance of a porcine collagen membrane was studied in a defect of the critical size left without grafting. The membranes used in bone regeneration have a smooth surface that must be in contact with the epithelial tissues and a rough surface that will be in touch with the grafted area to allow the anchoring of bone cells. Figure 11 corresponds to the SEM image of one of the membranes used. The rough surface is observed with numerous irregular areas that offer a potential anchorage to the bone cells that colonize the site during healing.

Figure 12 shows the results after applying F3 as a barrier in defects of critical size not grafted at 30 days, 60 days, and 90 days. Thirty days after application, it is observed that the empty defect remains without evidence of regeneration and is observed to be occupied by connective tissue for healing (Figure 12A). The presence of Collagen may correspond to the same membrane (Figure 12B). SEM analysis exhibited that the membrane uniformly covers the defect and appears to be integrated into the periosteum of the adjacent bone (Figure 12C, indicated with red arrows in the white circle).

After 60 days, the membrane appearance is fragmented with evidence of degradation. The fragments are immersed in the inflammatory infiltrate (Figure 12D,E) with an abundant presence of blood vessels (Figure 12F).

After 90 days of implantation, nuclei of bone formation are observed in the defect in the middle of a healing connective tissue (Figure 12G). At a higher magnification, fragments of the material are found immersed in a matrix of collagen fibers (Figure 12E).

The absence of new bone formation after 30 days indicates that the applied membrane cannot stimulate bone regeneration in a defect of critical size. For 60 days, the material is in the process of degradation, and there is a stimulus to the bone formation of blood vessels (angiogenesis). Collagen degradation would be expected to stimulate the healing process with the presence of new blood vessels and the construction of new bone nuclei.

In general, membranes are not expected to have a regenerative effect on bone defects, but they have an osteoconductive effect. They might also induce an osteopromoting effect to stimulate bone adhesion on the surface in contact with the defect [56]. The osteoconductive impact is also expected to help defect regeneration when a graft material is used [57].

The results found of bone neoformation after 60 days do not agree with the results shown by the groups Ramirez (2020) and Ferreira Bizelli (2022), who found early bone neoformation in defects of critical size in rat sealed membranes of porcine pericardium [58]. However, it must be considered that the design used was 8 mm in length, involving the sagittal suture, which may have influenced the results.

## 3. Materials and Methods

The three materials used in this study were two xenografts of bovine origin composed of anorganic granules of cancellous bone with a size of 0.25 to 1 mm (InterOss); a xenograft of bovine origin supplemented with Collagen in the form of a block composed of 90% anorganic granules of cancellous bone with a size of 0.25 to 1 mm and 10% collagen fibers (InterOss Collagen) and a collagen membrane developed from the bovine pericardium (InterCollagen Guide Extend). All the materials were manufactured by SigmaGraft (SigmaGraft Biomaterials. Fullerton, CA, USA).

We used twenty-four male Wistar rats (180 days old, 380 g) provided by the LABBIO laboratory Health Faculty of the Universidad del Valle in Cali, Colombia. The biomodels were randomly distributed in three groups of eight individuals for implantations of 30, 60, and 90 days, with three replicates for each sample in the three periods tested.

In each biomodel, two 5 mm bilateral preparations were made in parietal bone for 48 preparations. Four trials were organized; InterOss material was applied to the first; to the second, the InterOss Collagen material; the third was left without being implanted as an empty control, and the fourth was left without applying graft material but was covered with a collagen membrane (InterCollagen Guide).

For the research design, the recommendations of The ARRIVE guidelines (Animal Research: Reporting of in vivo Experiments) were followed [59] and were approved by the Ethics Committee with experimental animals of the Universidad del Valle (Cali, Colombia) through resolution No. 003 of 2020.

### 3.1. Characterization of the Xenografts

#### 3.1.1. Fourier Transform Infrared Spectroscopy (FT-IR)

Fourier transform infrared spectroscopy was used to identify the functional groups of the materials (F1, F2, and F3). An infrared (IR) spectrometer with attenuated total reflectance (ATR), PerkinElmer Spectrum Two model, was used in the transmittance mode. Spectra were obtained in the range from 4000 to 400 cm^−1^ at room temperature. The resolution scan was 4 cm^−1^, while the scanning speed was 1 cm/s, with 30 counts per sample.

#### 3.1.2. Thermal Analysis

A differential scanning calorimeter (DSC), Mettler-Toledo, model DSC 1 STAR^e^ System was used for the thermal analysis of the materials. For this purpose, approximately 4–7 mg of each sample were massed using an analytical balance, incorporated into aluminum capsules and introduced into the equipment. Once inside, the samples were heated under three temperature ramps, highlighting the heating, holding, and thermal cooling processes. The temperature range used for the analyses was from −30 °C to 300 °C, at a rate of 10 °C/min, under a nitrogen atmosphere, with a flow rate of 60 mL/min.

#### 3.1.3. X-ray Diffraction (XRD)

The characterization of the crystalline structure of the composites was evaluated by X-ray diffraction (XRD). The samples were analyzed in a Bruker diffractometer model D8 ADVANCE using Cu Kα1 (1.5406 Å) and Kα2 (1.5444 Å) radiation, in a spectral range: 2° < 2θ > 80°.

#### 3.1.4. Scanning Electron Microscopy (SEM)

To determine the surface morphology and qualitatively appreciate the porosity of the samples, they were supported on a copper adhesive conductive tape, then metalized with 15 nm of gold on their surface with the Cressington 108 auto Sputter Coater equipment to improve the conduction of the electrons. Finally, samples were placed in the Zeiss EVO MA 10 Scanning Electron Microscope (SEM), using the secondary electron detector SEI (Secondary Electron Image) at 20 kV. Chemical microanalysis was carried out with the energy-dispersive spectroscopy (EDS) probe (Model INCAPentaFETx3 instrument, Oxford Instruments, Abingdon, UK). 

### 3.2. Histological Analysis of Xenografts

#### 3.2.1. Surgical Preparations

Bilateral intraosseous preparations of 5 mm diameter and total thickness were made in parietal bones of the biomodels without involving bone sutures. First, the biomodels were weighed. Then, sedation using Ketamine 70 mg/kg (Blaskov Laboratory, Bogotá, Colombia) and Xylazine 30 mg/kg (ERMA Laboratories, Celta, Colombia). After that, trichotomy of the cranial surface, disinfection (Isodine solution^®^ from Sanfer laboratory, Bogotá, Colombia), infiltrative anesthesia (Lidocaine 2% with epinephrine, Newstetic, Guarne, Colombia). The intraosseous preparations used trephine of 5 mm external diameter (MIS Implants Bar-Lev Industrial Zone Misgav, 2015600, Israel).

After finishing preparations, the experimental materials were applied, and the defects were left empty as a control (with or without collagen membrane). The incision was closed with a 4-zero silk suture. Gentamicin 0.1% (Procaps, Barranquilla, Colombia) and Diclofenac 75 mg were applied intramuscularly (La Santé, Bogotá, Colombia) to the wound. After implantation, the biomodels were euthanized (sodium pentobarbital/sodium diphenylhydantoin, a dose of 0.3mL/kg of Euthanex, INVET Laboratory, Cota, Colombia).

#### 3.2.2. Sample Preparation

The recovered samples were fixed in buffered formalin for 48 h. One representative from each group was randomly selected for analysis by scanning electron microscopy and the other three for histological studies.

Histology samples were washed gently with PBS and processed in an Autotechnicon Tissue ProcessorTM (Leica Microsystems, Mannheim-Germany). They were included in paraffin blocks (Thermo ScientificTM HistoplastTM (Thermo Fisher Scientific, Waltham, MA, USA), cut in sizes of six micrometers with a Leica microtome, and staining was performed with the Hematoxylin-Eosin (H-E), Masson’s Trichrome (MT) and Gomori’s trichrome (GT) stains. Photomicrographs were taken with a Leica optical microscope, Leica DM750 microscope with a Leica DFC 295 camera, and Leica Application Suite version 4.12.0 (Leica Microsystem, Mannheim, Germany).

The samples selected for scanning electron microscopy studies were washed in phosphate buffer three times for five minutes and then dehydrated by immersion in ascending alcohols at 50%, 70%, 80%, 90%, and 100%, and dried at 37 °C for 36 h.

## 4. Conclusions

The three materials tested in our investigation showed in vivo biocompatibility and degradability. The two bone substitutes, InterOss (F1) and InterOss Collagen (F2) stimulated the regeneration of critical defects, evidencing an osteoconductive effect; however, the InterOss Collagen (F2) material presented earlier bone neoformation and had less bone remnant material upon completing the 90 days of implantation.

The InterCollagen Guide Extend membrane performed its barrier role. The empty defect remained without evidence of regeneration until 60 days and stimulated an angiogenesis process that could have influenced the partial neoformation observed at 90 days. Our results in this investigation demonstrated the promise of these xenografts as bone substitutes. In upcoming research, we will test the xenografts in preclinical assays and validate their use in human beings. 

## Figures and Tables

**Figure 1 molecules-27-05745-f001:**
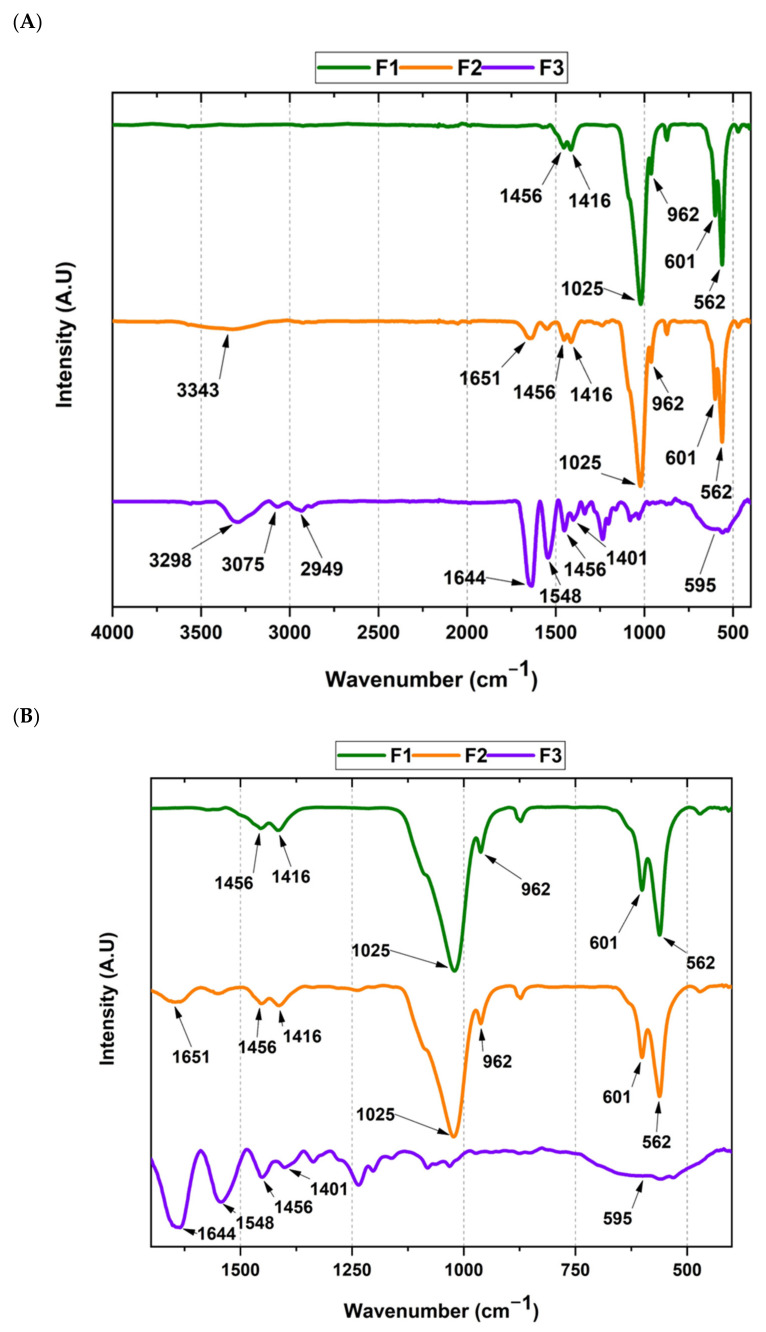
(**A**) FT-IR spectrum of the xenografts and collagen membrane. (**B**) Expanded 600–1700 cm^−1^ region of the FT-IR spectrum. F1: InterOss. F2: InterOss Collagen. F3: InterCollagen Guide.

**Figure 2 molecules-27-05745-f002:**
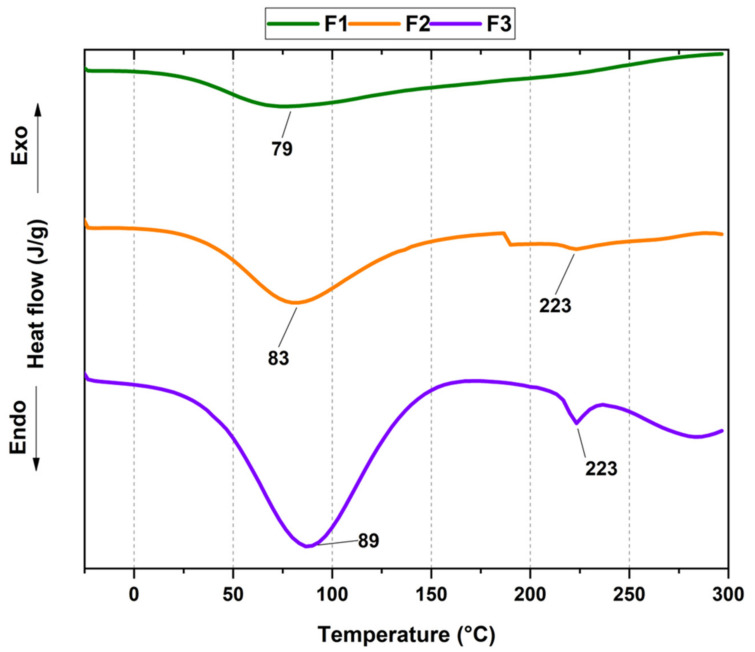
DSC thermal analysis of the composites.

**Figure 3 molecules-27-05745-f003:**
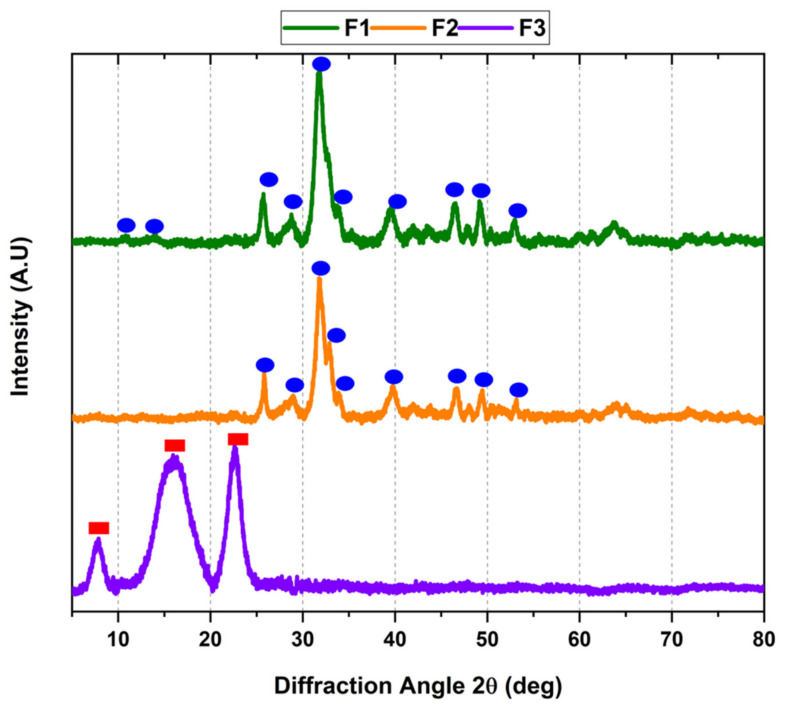
XRD diffraction pattern of F1, F2, and F3. F1: InterOss. F2: InterOss Collagen. F3: InterCollagen Guide.

**Figure 4 molecules-27-05745-f004:**
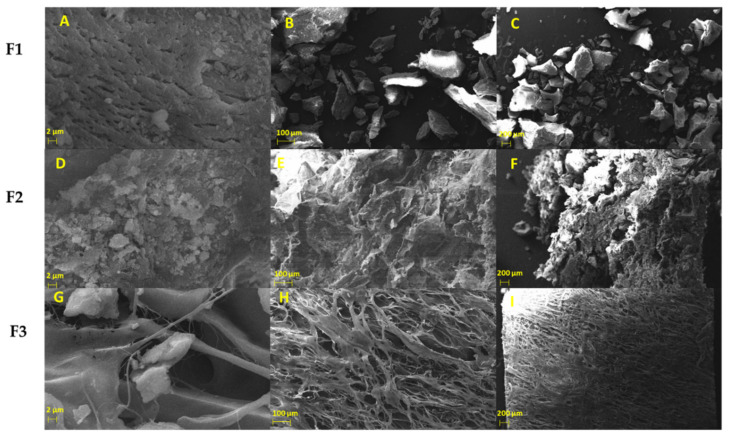
SEM analysis of F1 (**A**–**C**), F2 (**D**–**F**), and F3 (**G**–**I**). (**A**,**D**,**G**) at 5000×. (**B**,**E**,**H**) at 100×. (**C**,**F**,**I**) at 50×. F1: InterOss. F2: InterOss Collagen. F3: InterCollagen Guide.

**Figure 5 molecules-27-05745-f005:**
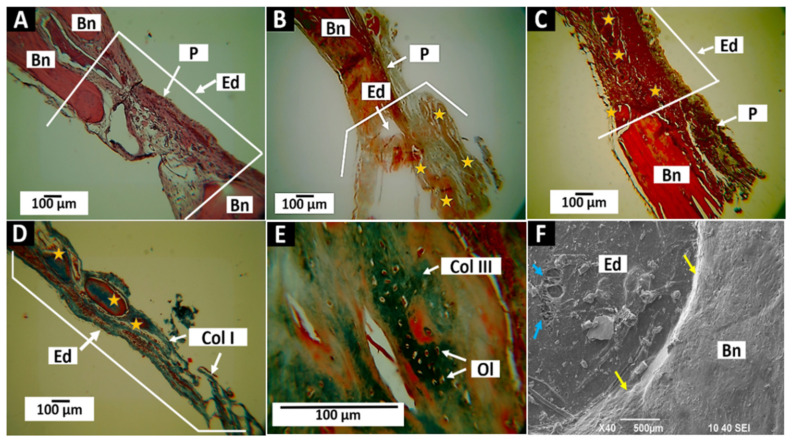
Healing of the control defect. (**A**): Preparation at 30 days at 10×. Hematoxylin—Eosin (HE) technique. (**B**): Preparation at 60 days at 10×. GT technique. (**C**): preparation at 90 days at 10×. HE technique. (**D**): Preparation at 90 days at 10×. Masson’s Trichrome (MT) and Gomori’s Trichrome (GT) (MT) technique. (**E**): Preparation at 90 days at 40×. GT technique. (**F**): Preparation at 30 days, SEM technique at 40×. Bn: bone. Ed: Empty defect. P: Periosteum. Stars: Nodules of bone formation. Yellow arrows: Interface preparation—Bone. Blue star: Unsealed or perforated areas. Col I: Type I Collagen. Col III: Type III collagen. Ol: Osteocytic lacunae.

**Figure 6 molecules-27-05745-f006:**
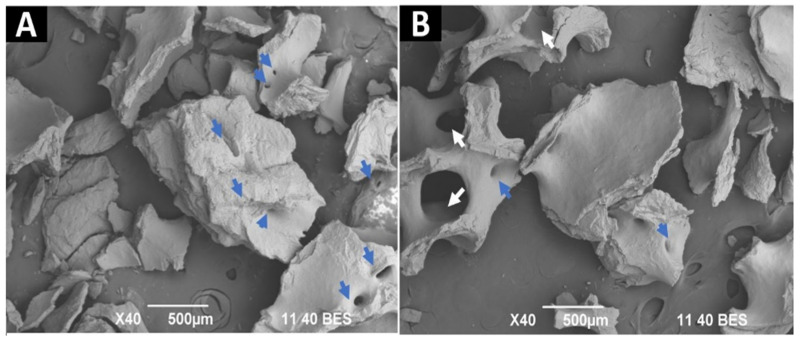
F1 (InterOss) SEM analysis. (**A**,**B**) 40×. Blue arrows: pores. White arrows: trabeculae with medullary cavities.

**Figure 7 molecules-27-05745-f007:**
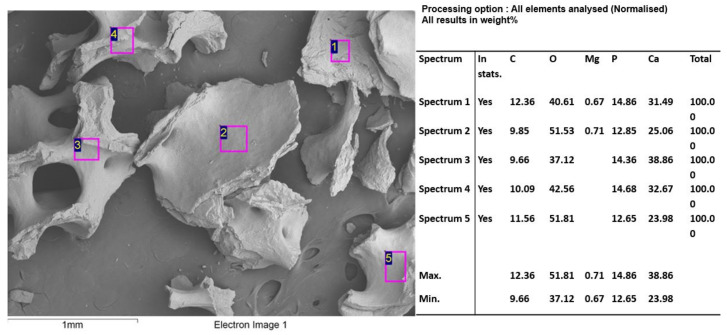
F1 (InterOss) SEM/EDS analysis.

**Figure 8 molecules-27-05745-f008:**
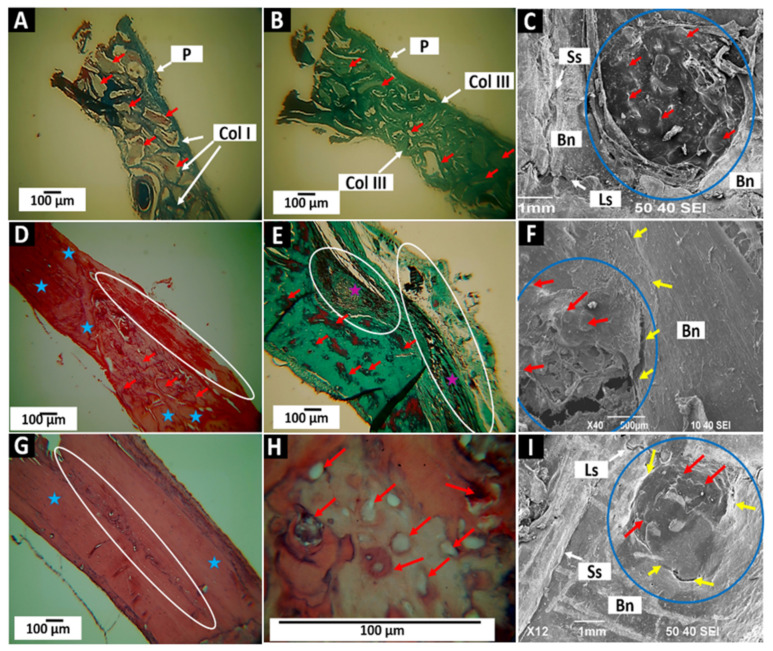
F1 material (InterOss) implanted in cranial bones. (**A**): Implantation at 30 days at 10×. Masson’s Trichrome (MT) technique. (**B**): Implantation at 30 days at 10×. Gomori’s (GT) technique. (**C**): Implantation at 30 days at 12×. SEM technique. (**D**): Implantation at 60 days at 10×. Hematoxylin—Eosin (HE) technique. (**E**): Implantation at 60 days at 10×. GT technique. (**F**): Implantation at 60 days at 40×. SEM technique. (**G**): Implantation at 90 days at 10×. HE technique. (**H**): Implantation at 90 days at 100×. HE technique. (**I**): Implantation at 90 days at 12×. SEM technique. P: Periosteum. Col I: Type I Collagen. Col III: Type III collagen. Ss: sagittal suture. Ls: lambdoid suture. Bn: bone. Blue stars: bone tissue in formation. Yellow arrows: Interface preparation—Bone. Red arrows: Remaining InterOss material. White oval: Areas with the remaining material. Blue circles: implantation zones.

**Figure 9 molecules-27-05745-f009:**
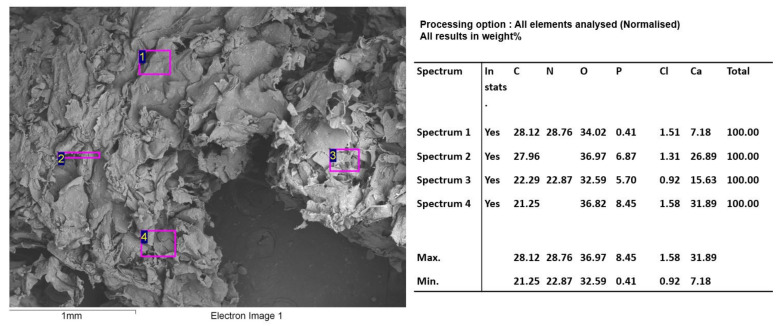
F2 (InterOss Collagen) SEM/EDS analysis.

**Figure 10 molecules-27-05745-f010:**
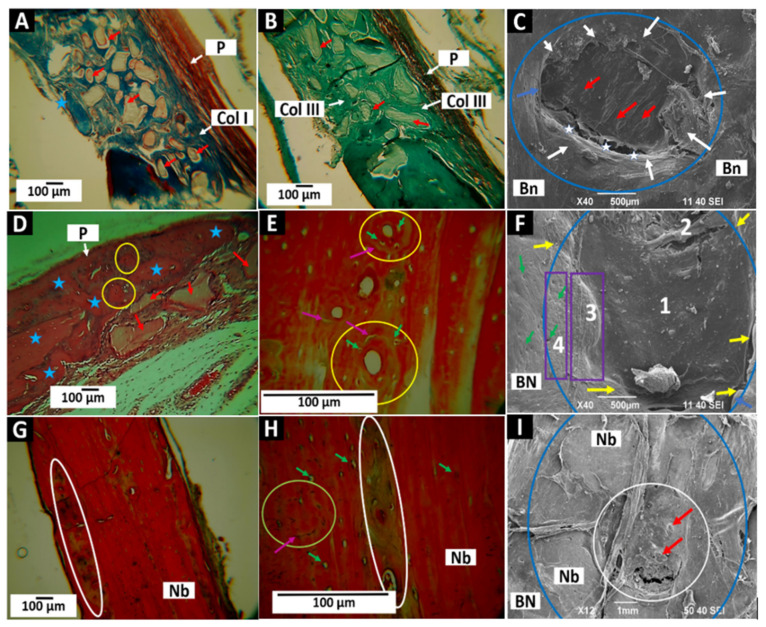
F2 (InterOss Collagen) material implanted in cranial bones. (**A**): Implantation at 30 days at 10×. MT technique. (**B**): Implantation at 30 days at 10×. GT technique. (**C**): Implantation at 30 days at 40×. SEM technique. (**D**): Implantation at 60 days at 10×. HE technique. (**E**): Implantation at 60 days at 10×. GT technique. (**F**): Implantation at 60 days at 40×. SEM technique. (**G**): Implantation at 90 days at 10×. HE technique. (**H**): Implantation at 90 days at 40×. HE technique. (**I**): Implantation at 90 days at 12×. SEM technique. P: Periosteum. Col I: Type I collagen. Col III: Type III collagen. Bn: bone. Blue stars: bone tissue in formation. Blue arrows: Remaining area of the preparation. Yellow circles: “osteons-like.” Green Arrows: Osteocytic lacunae. Purple arrows: Cement lines. Red arrows: Remaining InterOss material. Nb: Newly formed bone. White oval: Areas with the remaining material. Blue circles: implantation zones. 1–3: Areas with different degrees of bone neoformation.

**Figure 11 molecules-27-05745-f011:**
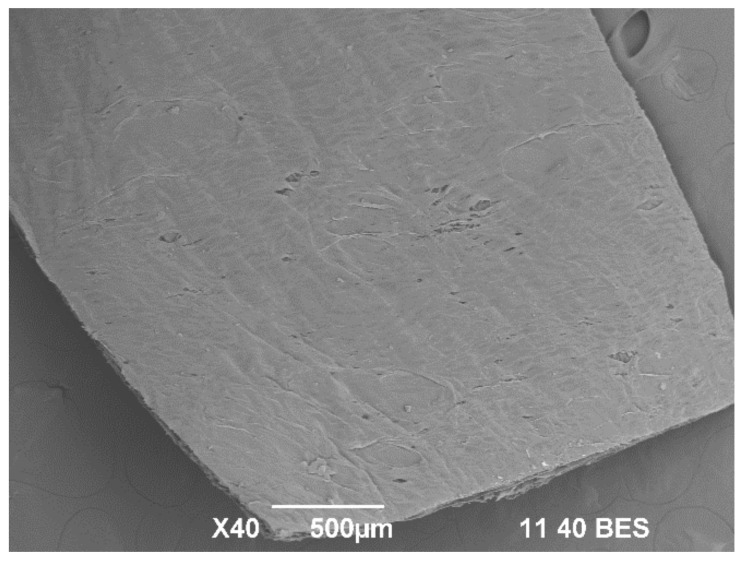
F3 (InterCollagen Guide) membrane SEM analysis.

**Figure 12 molecules-27-05745-f012:**
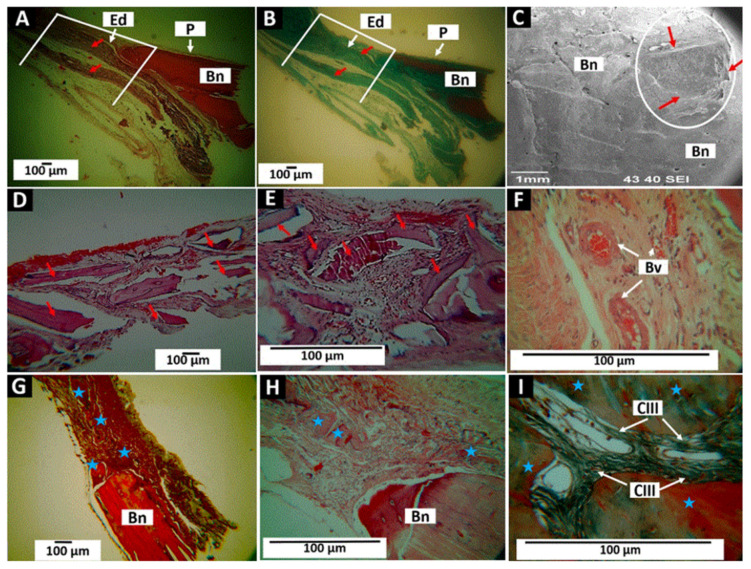
F3 (InterCollagen guide Extend membrane) material applied in intraosseous defects in cranial bones. (**A**): Implantation at 30 days at 4×. HE technique. (**B**): Implantation at 30 days at 4×. GT technique. (**C**): Implantation at 30 days at 40×. SEM technique. (**D**): Implantation at 60 days at 10×. HE technique. (**E**): Implantation at 60 days at 10×. GT technique. (**F**): Implantation at 60 days at 100×. HE technique. (**G**): Implantation at 90 days at 4×. HE technique. (**H**): Implantation at 90 days at 40×. HE technique. (**I**): Implantation at 90 days at 100×. HE technique. P: Periosteum. Ed: Empty defect. Bn: remaining bone. Bv: blood vessel. CIII: Collagen type III. Red arrows: Material. Blue stars: new bone formed.

## Data Availability

Data from the experiments are available upon request from the corresponding author.

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
