# Peer review of "Comparison of Two Bovine Commercial Xenografts in the Regeneration of Critical Cranial Defects"

_molecules, 2022, doi:10.3390/molecules27185745_

Round 1
Reviewer 1 Report
The manuscript Comparison of two bovine commercial xenografts in the regeneration of critical cranial defects submitted to Molecules - Manuscript Number: molecules-1871567 describes characterization of two bovine commercial xenografts and a collagen membrane.
Some remarks have to be taken into account by the authors:
- In Figure 1, spectra are blurry. The legend is hardly legible, increase the font size and change the line color. Please correct.
- IR spectra in the region 600-1700 cm-1 should be zoomed.
- In Figures 2 and 3, the legend is hardly legible, increase the font size and change the line color. Please correct
- In sec. 2.1.1 to 2.1.3, the entire discussion is quite superficial and needs to be improved with critical discussions about the differences and similarities e.g. Why does the endothermic dehydration transition have different values for F1, F2 and F3 samples?
- How many samples of one type (or the number of replicates of the experiment) were used in the studies?
- Sufficient details of used techniques should be described to allow others to replicate and build on published results e.g. FTIR analysis i.e. the measurement conditions such as the resolution and the number of scans.
Author Response
We thank you for all the valuable suggestions from the reviewer. Carefully, we try to address point by point of them. You can find our answers in the attached response letter and the corrected version of the manuscript.
Reviewer 1
The manuscript Comparison of two bovine commercial xenografts in the regeneration of critical cranial defects submitted to Molecules - Manuscript Number: molecules-1871567 describes the characterization of two bovine commercial xenografts and a collagen membrane.
Some remarks have to be taken into account by the authors:
- In Figure 1, spectra are blurry. The legend is hardly legible; increase the font size and change the line color. Please correct.
R// We are deeply thankful for the recommendation from the reviewer. We increased the font size and changed the colors of the figure.
- IR spectra in the region 600-1700 cm-1 should be zoomed.
R// We are deeply thankful for the recommendation from the reviewer. We zoomed in on the indicated region.
- In Figures 2 and 3, the legend is hardly legible, increase the font size and change the line color. Please correct
R// We are deeply thankful for the recommendation from the reviewer. We increased the font size and changed the colors of the figures.
- In sec. 2.1.1 to 2.1.3, the entire discussion is quite superficial and needs to be improved with critical discussions about the differences and similarities e.g. Why does the endothermic dehydration transition have different values for F1, F,2, and F3 samples?
R// We are deeply thankful for the suggestion from the reviewer. In section 2 (lines 110-112) and subsections 2.1.1 to 2.1.3, more discussion was introduced, with a comparison between formulations (highlighted in blue).
- How many samples of one type (or the number of experiment replicates) were used in the studies?
R// We are thankful for the correction from the reviewer. We added to line 543: “with three replicates for each sample in the three periods tested.”
- Sufficient details of used techniques should be described to allow others to replicate and build on published results e.g. FTIR analysis i.e. the measurement conditions such as the resolution and the number of scans.
R// We are deeply thankful for the recommendation from the reviewer. The 2.2 section "characterization of the xenografts" was rewritten carefully, adding enough details. Some equipment was corrected since we had some erroneous information.

Reviewer 2 Report
The study by Valencia-Llano et al. investigates the effect of bovine xenografts for regeneration of critical sized bone defects in rats. The manuscript reads fairly well, but some sentences and paragraphs are worded a bit oddly and could be rephrased for clarity. Below are some specific comments to improve the manuscript.
Specific comments:
-
FTIR should be introduced before being abbreviated in the abstract.
-
Since rats do not have true Havarisian remodeling, another word than “osteon” would be more appropriate to use.
-
Consider rephrasing this sentence for clarity: “It has been considered antigenic potential, and some of the qualities they can present for coming from a living being are lost due to the extraction process…
-
® can be omitted in scientific writing.
-
The two last sentences in the Introduction starts with “then” - consider an alternative word to make it more engaging.
-
Please clarify why the commercial xenografts were scrutinized to carefully with FT-IR, DSC, and SEM.
-
A sample size calculation to justify the use of 8 animals pre group is mandatory.
Author Response
We thank you for all the valuable suggestions from the reviewer. Carefully, we try to address point by point of them. You can find our answers in the attached response letter and the corrected version of the manuscript.
Reviewer 2
The study by Valencia-Llano et al. investigates the effect of bovine xenografts for regeneration of critical-sized bone defects in rats. The manuscript reads fairly well, but some sentences and paragraphs are worded a bit oddly and could be rephrased for clarity. Below are some specific comments to improve the manuscript.
Specific comments:
FTIR should be introduced before being abbreviated in the abstract.
R// We are deeply thankful for the recommendation from the reviewer. The proper corrections were introduced in the abstract sections (lines 7 and 11)
Since rats do not have true Havarisian remodeling, another word than "osteon" would be more appropriate to use.
R// We are deeply thankful for the recommendation from the reviewer. We changed the word "osteon" to "osteon-like."
Consider rephrasing this sentence for clarity: "It has been considered antigenic potential, and some of the qualities they can present for coming from a living being are lost due to the extraction process…
R// We are deeply thankful for the recommendation from the reviewer. We rewrote the sentence (lines 50-51): "However, they might introduce an antigenic response, and some native properties might be lost due to the chemical extraction."
® can be omitted in scientific writing.
R// We are deeply thankful for the recommendation from the reviewer. All the symbols ® were removed from the manuscript.
The two last sentences in the Introduction starts with "then" - consider an alternative word to make it more engaging.
R// We are deeply thankful for the recommendation from the reviewer. We rewrote the final sentences: "Thus, bone regeneration will be attributed to the xenografts and not to a spontaneous healing process."
Please clarify why the commercial xenografts were scrutinized to carefully with FT-IR, DSC, and SEM.
R// We thank the recommendation. Although the xenografts are commercially available, we must probe their main chemical and crystalline properties, as we reported in our previous paper, and link them with their in vivo response. FT-IR thoroughly characterized the composites to demonstrate the presence of the main functional groups of hydroxyapatite and collagen. For its part, the analysis of the thermal transitions by DSC and the XRD study allowed a better understanding of the crystalline properties of the materials and to relate them to the biodegradation/reabsorption capacity in the biomodels. Finally, the analysis of the microstructure by SEM allowed us to evidence material's porosity and morphology and their relationship with the biodegradation/reabsorption in the biomodels bone tissue.
A sample size calculation to justify the use of 8 animals per group is mandatory.
R// We thank the comment from the reviewer. However, in this type of test that involves animals, no sample size calculation is performed because the "Reduction" principle of Russel and Burch ("The Principle of Humane Experimental Technique") must be applied. In place of sample calculation, ISO 10993-5 8 TEST PROCEDURE, section 8.1 "Number of replicates" says that "a minimum of three replicates shall be used for assay samples and controls." In this case, each group corresponded to periods of 30, 60, and 90 days, and in each group, three products and one control were tested. As the critical size defect model of five millimeters allows two preparations to be made in each biomodel, it was only necessary to use eight biomodels per group, fulfilling the three replicates and leaving some samples available for SEM analysis.
